# Radiometric Assessment of ICESat-2 over Vegetated Surfaces

Amy Neuenschwander [1,*], Lori Magruder [1,2], Eric Guenther [3], Steven Hancock [4] and Matt Purslow [4]

1. Center for Space Research, University of Texas at Austin, Austin, TX 78759, USA; lori.magruder@austin.utexas.edu
2. Department of Aerospace Engineering, University of Texas at Austin, Austin, TX 78712, USA
3. Applied Research Laboratories, University of Texas at Austin, Austin, TX 78758, USA; eguenther@arlut.utexas.edu
4. School of GeoSciences, University of Edinburgh, Edinburgh EH9 3FF, UK; steven.hancock@ed.ac.uk (S.H.); matthew.purslow@ed.ac.uk (M.P.)
* Correspondence: amy@csr.utexas.edu; Tel.: +1-512-461-1884

**Abstract:** The ice, cloud, and land elevation satellite-2 (ICESat-2) is providing global elevation measurements to the science community. ICESat-2 measures the height of the Earth's surface using a photon counting laser altimeter, ATLAS (advanced topographic laser altimetry system). As a photon counting system, the number of reflected photons per shot, or radiometry, is a function primarily of the transmitted laser energy, solar elevation, surface reflectance, and atmospheric scattering and attenuation. In this paper, we explore the relationship between detected scattering and attenuation in the atmosphere against the observed radiometry for three general forest types, as well as the radiometry as a function of day versus night. Through this analysis, we found that ATLAS strong beam radiometry exceeds the pre-launch design cases for boreal and tropical forests but underestimates the predicted radiometry over temperate forests by approximately half a photon. The weak beams, in contrast, exceed all pre-launch conditions by a factor of two to six over all forest types. We also observe that the signal radiometry from day acquisitions is lower than night acquisitions by 10% and 40% for the strong and weak beams, respectively. This research also found that the detection ratio between each beam-pair was lower than the predicted 4:1 values. This research also presents the concept of ICESat-2 radiometric profiles; these profiles provide a path for calculating vegetation structure. The results from this study are intended to be informative and perhaps serve as a benchmark for filtering or analysis of the ATL08 data products over vegetated surfaces.

**Keywords:** ICESat-2; ATLAS; ATL08; laser altimetry; radiometry

## 1. Introduction

In 2018, NASA launched the space-based laser altimetry mission ice, cloud, and land elevation satellite-2 (ICESat-2). This mission was the result of the priority recommendation of the National Research Council Decadal Survey in 2007 to dedicate a mission to continue elevation time series in the cryosphere. Beyond the primary scientific goal, however, the ICESat-2 system facilitates measurements over other surface types that contribute to a broad spectrum of science disciplines. The onboard instrument ATLAS (advanced topographic laser altimetry system) uses a single 532 nm laser with a diffractive optical element that splits the laser into six beams organized as three beam pairs [1,2]. Each beam pair is approximately 3 km apart in the across track direction and consists of a strong beam and weak beam of relative energy. The detection ratio between the strong and weak beam was designed to be 4-to-1. Depending upon the surface reflectance and atmospheric conditions, up to 16 photons per outgoing shot could be detected on the $4 \times 4$ detector array for the strong beam and 4 photons on the $2 \times 2$ detector array for the weak beam. These maximum photon detections are indicative of saturation conditions until the detector satisfies the dead-time recovery period. The size of each ICESat-2 footprint on the ground has been determined to range between 10–12 m in diameter [3,4].

What sets the ATLAS instrument apart from other laser ranging systems is that it uses a photon-counting detection technology, which enables a detection modality sensitive to individual photons. A benefit of the photon counting system is that the laser can operate at a high repetition rate, thus improving the along-track measurement resolution. The small footprint and the high laser repetition rate results in a near-continuous along-track sampled profile. ICESat-2, specifically, has a 10 kHz laser repetition rate combined with the altitude and velocity of the satellite, resulting in a 70 cm separation for each shot on the Earth's surface in the along-track direction.

Because ICESat-2 is a photon counting system rather than a full-waveform system, only a few photons are detected by the ATLAS receiver for each outgoing shot. The ATLAS detector detects both signal photons (i.e., reflected from the surface) and background noise photons. The number of detected photon events, both signal and background, associated with each outgoing laser pulse is a function of the laser wavelength, transmitted laser energy, surface reflectance, solar conditions, and scattering and attenuation in the atmosphere. For measurements over vegetation, the number of signal photons detected is distributed between canopy top, mid-canopy, and terrain. The return photon distribution is a function of the reflectance of both the ground and canopy [5] while also dependent on the vegetation structure [6]. Based on early results from [6,7], general trends for the number of photon counts per laser shot were observed while analyzing residuals of ICESat-2 terrain heights compared to airborne lidar terrain heights. In [7], it was observed that the signal return strength varied between night versus day acquisitions and with snow versus no-snow conditions providing a basis for a more in-depth exploration regarding how solar elevation angle and seasonal variations might impact the signal radiometry. One goal of this study was to better understand the influence laser energy, solar elevation, and atmospheric scattering have on the number of detected signal photons.

Additionally, this study explores the radiometric performance of the ATLAS instrument over different forest types. Generally speaking, radiometry is defined as the number of signal photons detected per outgoing laser shot and, thus, we followed the general assumption that the ability to accurately detect the surface and estimate its height (whether canopy or terrain) was dependent upon a sufficient number of photons reflected from that surface. To determine the influence of radiometry on the estimated terrain and canopy heights from the ATL08 (land and vegetation) data product, the radiometry must be calculated as a function of either ground or canopy. Here, these radiometric calculations were made possible by utilizing the photon labels for each surface type provided on the ATL08 data product.

*ATLAS*

The ATLAS instrument collects reflected light (532 nm) through a 0.8 m telescope via a narrow bandpass (30 pm) filter to the detector [8]. Photons for each beam are passed into a photomultiplier tube to convert the photon event into a digital event which is time-tagged and used for further analysis. A diffractive optical element (DOE) splits the ATLAS outgoing laser into six beams with an energy ratio of approximately 4:1 between the strong and weak beams. In terms of nomenclature, the beams are numbered such that 1, 3, and 5 are the strong beams and 2, 4, and 6 are the weak beams [1].

The beam numbering is constant over time regardless of the satellite orientation. This nomenclature differs from the ground-track left-right (e.g., GT1L or GT1R) naming convention as the satellite yaw position determines which beam occupies which track position. Beams 1, 3, and 5 are the strong beams and beams 2, 4, and 6 are weak as shown in Figure 1. The detector for each of the three strong beams consist of a $4 \times 4$-pixel array (16 pixels), whereas the detector for each weak beam is a $2 \times 2$ array (4 pixels) [8]. Thus, the strong beam detector could potentially record 16 photons across the array before saturation for a given pulse of emitted laser energy. Saturation for the weak beam optics occurs at 4 received photons. The focal plane array design configuration accommodates the relative energy levels of the strong and weak beams. Similarly, the outgoing energy of the strong

beams are four times the outgoing energy of the weak beams [9]. During the on-orbit commissioning phase of the satellite, the transmit laser energy was adjusted such that the strong beams of ATLAS were detecting on average 9 signal photons per pulse over the highly reflective ice sheets [10]. In [9] they also report strong beam radiometry rates over Antarctica of approximately 9 photons per shot for beams 1 and 5 and 7.5 photons per shot for beam 3. They also report weak beam radiometry values of 2.25 which yields the expected 4:1 energy ratio. This photon count criterion was determined to be optimal rate for cryospheric studies during pre-launch analysis. Thus far, the on-orbit radiometry over land ice has maintained the value between 8–9 photons/shot for the strong beams and approximately 2 photons/shot for the weak beams over Antarctica [10].

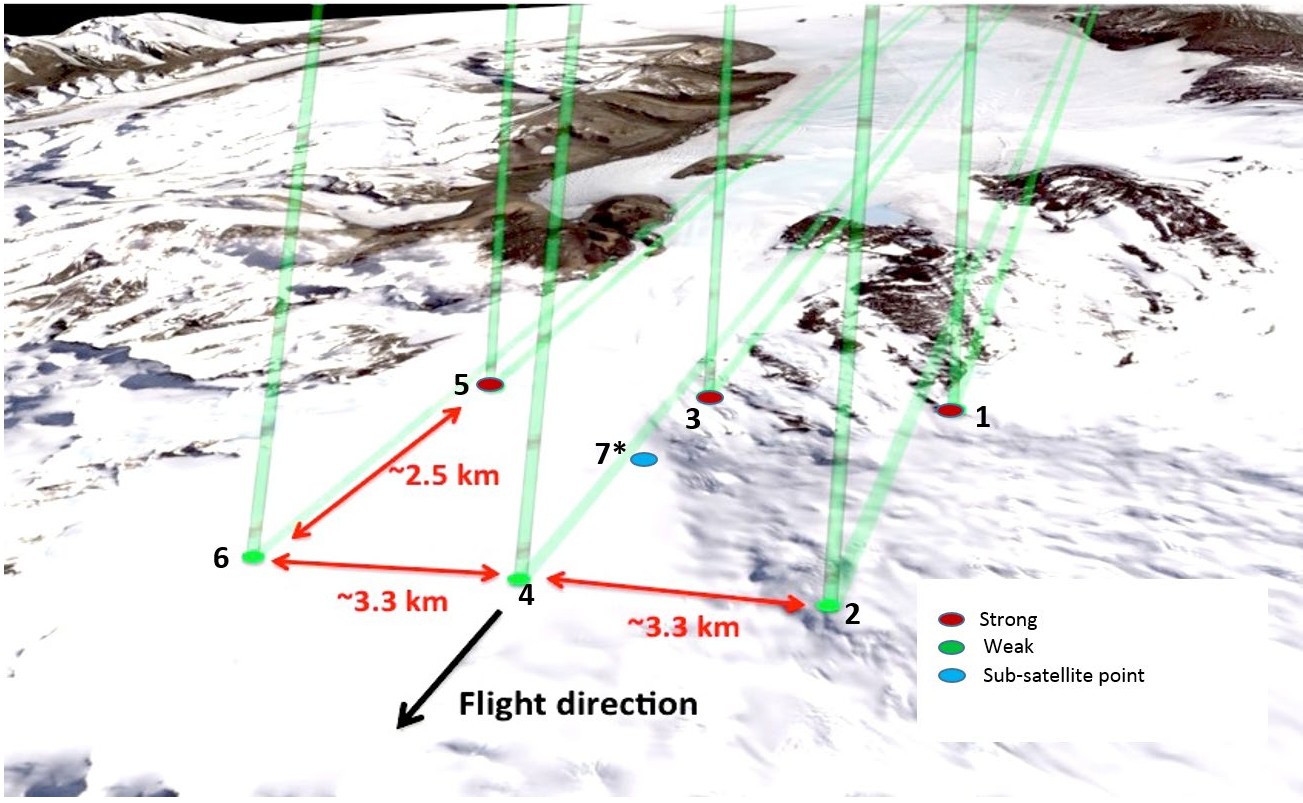

**Figure 1.** Beam configuration for ATLAS instrument on ICESat-2. Beams 1, 3, and 5 are strong beams and 2, 4, and 6 beams have a fourth of the energy. Spot 7 on the diagram is a virtual point representative of the sub-satellite point.

Over land, pre-launch studies were estimated, and early on-orbit analysis proved that the laser energy settings optimized for ice sheet reflectance would provide roughly one photon per pulse from both the ground and vegetation strong beam returns due to the lower reflectance. The actual number or expected returns, however, has been shown to vary for different forest types [11]. This research intends to quantify the radiometry for different forest types to understand the mechanisms associated with signal interpretation, system response, and achievable science quality applications with photon counting technology. In 2016, [5] reported the photon rates expected for boreal, temperate, and tropical forests based on design cases provided by the ICESat-2 project science office (Table 1). Prior to launch, the values listed for these design cases provided an early understanding and means for simulating the likely radiometric response of ICESat-2 over vegetated surfaces. Recently, [6] reported strong beam ground and canopy rates radiometry for Canadian boreal forest of 1.03 and 1.19 photons/shot, respectively.

**Table 1.** Pre-launch design cases and photon numbers provided by the ICESat-2 Project Science Office [5].

| Vegetation Type | Poisson Mean Photoelectrons Per Outgoing Shot | Poisson Distribution |
|---|---|---|
| Boreal Forest | 1.0 | [0 1 2] |
| Temperate Forest | 1.9 | [0 1 2 3] |
| Tropical Forest | 0.6 | [0 1] |

The radiometry for ICESat-2 is defined as the number of reflected signal photons divided by the number of outgoing laser shots. Each signal photon captured by ATLAS includes the transmit time, which allows users to determine the radiometry over different land surface types. Based on the photon classification results provided on the ATL08 data product, the radiometry is further partitioned for canopy returns and ground returns. As this research aims to examine the on-orbit radiometry over vegetated surfaces, we posited the following questions.

(1) What is the overall observed radiometry for three forest types used as pre-launch design cases: boreal, temperate, and tropical?
(2) What is the ATLAS beam radiometric observed variability?
(3) What impact does atmospheric scattering have on the observed radiometry
(4) What impact does day **vs.** night (e.g., solar elevation) have on the observed radiometry?
(5) What impact does snow have on the observed radiometry?

## 2. Materials and Methods

### 2.1. Study Areas

To answer our research questions, we examined the full ICESat-2 time-series (release 004) over several forested regions globally (Figure 2), including Finland (fully country extent; ~130,000 km$^2$), Alberta Canada (110–120W, 54–60N; ~739,250 km$^2$), Tapajos Brazil (0–6 S, 50–56W; ~443,500 km$^2$), Congo (1–3 S, 19–23 E; ~147,800 km$^2$), mid-South USA: Kentucky/Tennessee (35–38.6 N, 81.7–88.4 W; ~288,300 km$^2$), and central Europe: Germany, Czech Republic, and Poland (8–18 E, 48–53 N; ~616,000 km$^2$). These sites constitute a broad sampling of three forest types: boreal forest (Finland and Alberta), temperate forest (central Europe and mid-South USA), and tropical forest (Tapajos and Congo). All ICESat-2 ATL08 data are from release 004 (rel004) and cover the time period from October 2018 through February 2021 and are available from the National Snow and Ice Data Center [12].

### 2.2. ATL08 Data Filtering

For this analysis, the ATL08 data were initially filtered to reject those ATL08 segments with canopy heights (h_canopy) greater than 100 m. These filtered segments were attributed to photons from low lying clouds that precluded adequate retrievals from the canopy and terrain. Since this study examined large geographic extents and the true heights of the tallest trees within the area are unknown, we used a canopy height maximum of 100 m, which was arbitrary. Certainly, this initial canopy height filter can be reduced when working in areas where the maximum tree height is well known as a means to reject truly errant ATL08 segments. A second initial filter on the ATL08 segments was based on eliminating those segments with radiometric parameter values that exceed 16 photons per shot. The ground and canopy radiometric parameter on the ATL08 data product was (photon_rate_te) and (photon_rate_can), respectively. A strong beam ATL08 segment that contained a total radiometric value higher than 16 was likely indicative of cloud corruption in the data or over standing water as the ATLAS detector can only detect 16 photons per outgoing shot before saturation. An additional parameter on the ATL08 data product that was helpful for rejecting likely errant ATL08 segments was the (h_dif_ref) parameter, which represented the height difference between the ICESat-2 estimated ground surface and the reference

DEM used by the ICESat-2 ground systems. For Release 004 of ICESat-2, the reference DEM was the MERIT DEM over non-polar regions. The MERIT DEM was derived from the SRTMv3 and AW3D products and strived to provide a surface model absent of height artifacts from absolute bias, speckle noise, and vegetation [13]. For this analysis, we rejected ATL08 segments where the absolute height difference (h_dif_ref) was greater than 30 m.

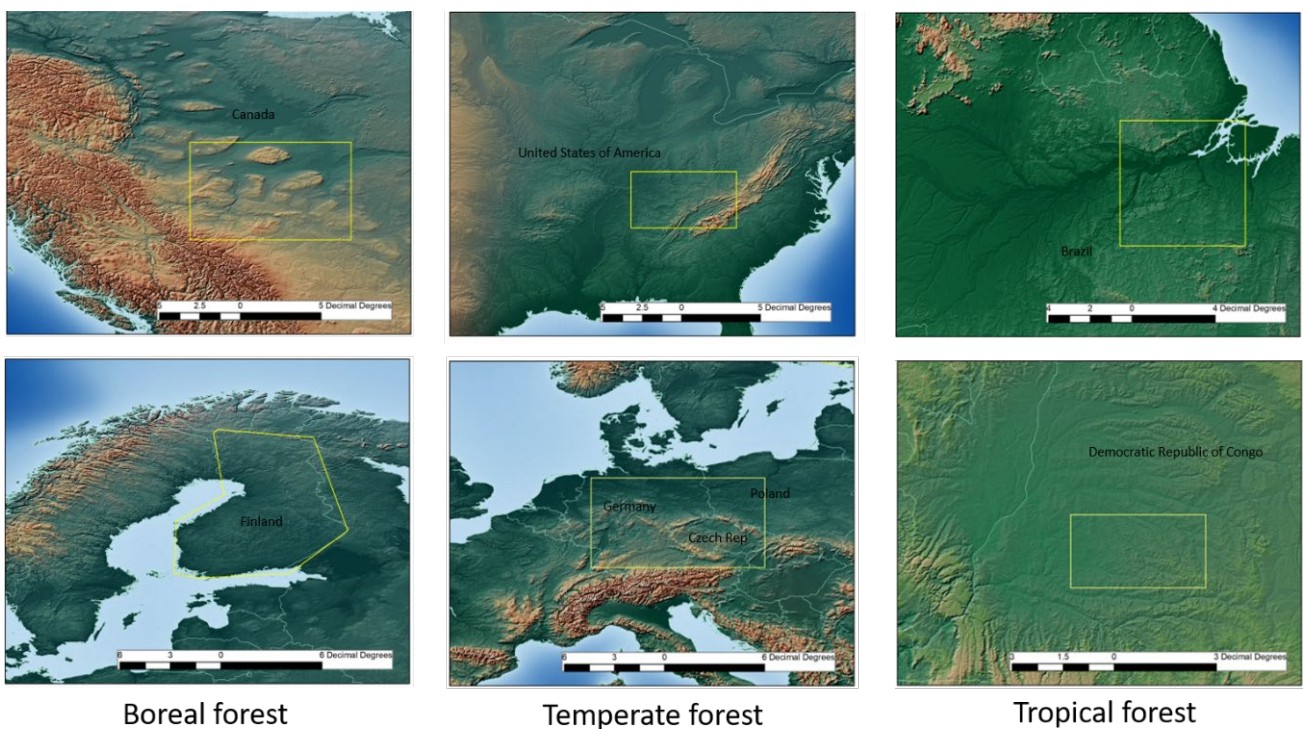

**Figure 2.** Study areas used in this research representing three general forest types: boreal, temperate, and tropical.

Once the data were aggregated based on the aforementioned rejection parameters, we generated a series of indices allowing us to filter or further stratify the data based on beam number, solar background conditions (i.e., day versus night), atmospheric scattering, landcover, and presence/absence of snow or ice in each ATL08 segment. The beam number was found in the gtx/group in the ATL08 hdf5 file and is listed as (atlas_spot_number). As described previously, beams 1, 3, and 5 were the strong beams and beams 2, 4, and 6 were the weak beams. To discern day versus night, we used the solar_elevation parameter with positive values representative of day acquisitions and negative elevations representative of night acquisitions. To determine a presence of atmospheric scattering, we utilized the multiple scatterer warning (msw_flag) on the ATL08 data product. This flag was calculated from the ICESat-2 atmospheric data product, ATL09. A msw_flag value of 0 indicated no observed scattering in the atmosphere. A msw_flag > 0 indicated the presence of scattering or cloud layers at different levels within the vertical column of the atmosphere [14]. For our purposes, we considered any msw_flag value > 0 as scattering that might affect the ability to accurately resolve the surface. Moreover, in this analysis we used landcover value provided on the ATL08 data. For release 004, the segment landcover corresponds to the MODIS global landcover comprised of 17 land cover classes. In this analysis, we selected landcover associated with forests: 1 = evergreen needleleaf, 2 = evergreen broadleaf, 3 = deciduous needleleaf, 4 = deciduous broadleaf, 5 = mixed forest, 6 = closed shrublands, and 8 = woody savanna. The woody savanna class was included as it also corresponds to sparse/open sections of boreal forest. Finally, a snow flag on the ATL08 data product was provided to help users determine the likelihood of snow-free data. The snow flag was derived from the

NOAA snow-ice daily product using a value of 0 to indicate ice-free water, 1 for snow-free land, 2 for a presence of snow, and 3 a presence of ice.

### 2.3. ATL08 Radiometry

Since the ATL08 algorithm determines photon labels for ground, canopy, top of canopy, and noise [15], a radiometric rate can be determined for ground photons per shot or canopy photons per outgoing laser shot. On the ATL08 data product, the radiometric rate was defined as the number of photons (canopy or ground) detected per outgoing shot averaged within the 100 m window for each ATL08 segment. During preparation of this manuscript, we discovered an error in the calculation of the radiometric rates reported on the ATL08 data product. Furthermore, the radiometric rates reported on the ATL08 data product were not adjusted for background noise. For our analysis, we calculated the true radiometric rates based on the labeling of ATL08 photons. Furthermore, we removed the impact of background noise on our radiometric counts over the forest canopy. Since the ground had a small spread of photons we did not attempt to adjust the number of ground photons for background noise. To calculate the number of background noise photons in our count of canopy photons for each 100 m ATL08 segment, we utilized the backgrd_counts_reduced and backgrd_in_height_reduced parameters [8,16] from ATL03 data products:

$$N = h_{canopy} * \left( \frac{background\_counts\_reduced}{backgrd\_in\_height\_reduced} \right) \tag{1}$$

where $N$ is simply the number of noise photons that we removed from our counting of canopy photons in each ATL08 segment prior to calculating the canopy radiometric rate and $h_{canopy}$ represents the 98% relative canopy height within each ATL08 segment. Since the background noise rate during night acquisitions is quite low [16], we found that the background noise correction did not reduce our canopy radiometric rate at night.

An illustration of the ATL08 labeled photons for each forest type is shown in Figure 3 below. Each acquisition highlighted in Figure 3 corresponds to a day acquisition. These cases are easily recognizable due to the significant solar background (noise points) observed both above and below the surface. In each of these profiles, yellow dots are representative of ground photons, dark green dots are representative of canopy photons, and light green dots are representative of top of canopy photons.

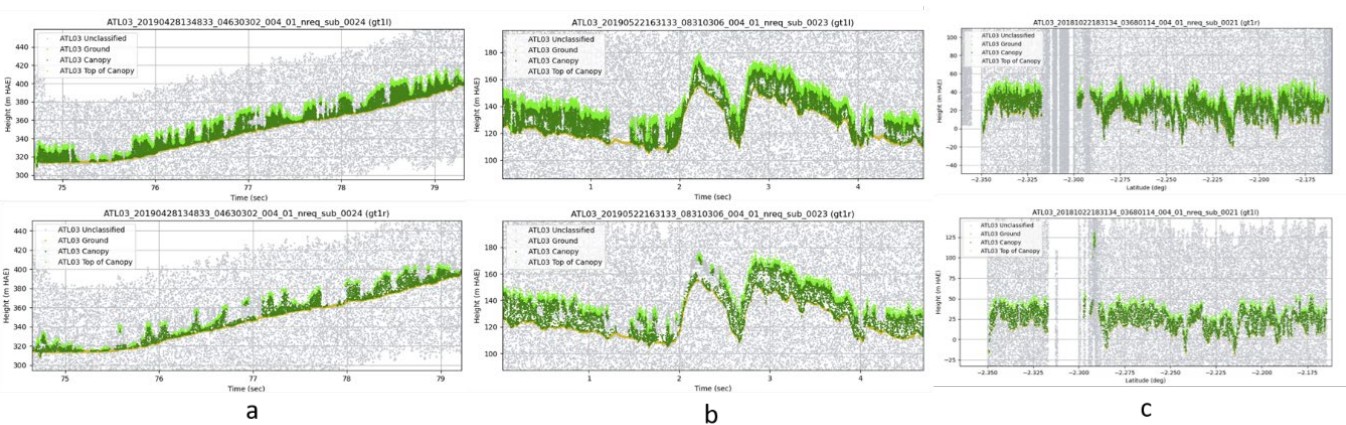

**Figure 3.** Profiles of the labeled photons from the ATL08 algorithm for three different forest types; (**a**) Alberta: boreal forest, (**b**) Germany: temperate forest, and (**c**) Tapajos: tropical forest. The strong beams are on the top row and weak beams are on the bottom row.

## 3. Results

### 3.1. Boreal Forests

Figure 4 below presents the radiometric histograms for boreal forest over Alberta, Canada for both the strong (top row) and weak (bottom row) beams. The histograms were

generated based on the ATL08 photon classifications to allow a comparison between the ground and canopy radiometries. Thus, the ground radiometry was the number of labeled ground photons per outgoing shot averaged across each 100 m ATL08 segment. Similarly, the canopy radiometry indicated the number of canopy photons per outgoing shot averaged across each 100 m ATL08 segment. The total radiometry reflects combined ground and canopy (or total signal) radiometric values. The radiometric histograms provide insights for the conditions explored (day versus night, clear versus cloud) but are solely focused on no snow scenarios. As previously mentioned, a positive scattering flag value was interpreted in this situation as the presence of atmospheric scattering or cloud layers. We took the opportunity to label these cases as "cloud". Looking at the results for the strong beam (top row), "cloudy-day" histogram showed the lowest total radiometric value. The distributions for the "clear-day", "clear-night", and "cloudy-night" all had a similar distribution of photons per segment, but "clear-night" had the highest mode centered near a value of 2.0 in the total radiometry. The distributions for the strong-beam ground and canopy radiometry show similar patterns and the mode of each distribution was approximately 1 and 0.7, respectively. When considering the weak beams (bottom row), there was a clear separability between the day and night acquisitions with the night total radiometric values being nearly double those observed during the day. The distribution of total radiometry in the weak beam at night was centered at 1.2 photons/shots, whereas the mode of the day total radiometry was centered near 0.6. In daytime acquisitions, the distribution of "cloud" segments radiometric properties for both ground and canopy appear to be impacted in the strong beam (in that they are shifted to the left). This implied that scattering in the atmosphere reduced the observed radiometry compared to "clear" sky conditions. This pattern, however, was not observed with the weak beam. For the weak beam cases, the radiometric distributions of both canopy and ground photons have similar shapes and modes regardless of atmospheric scattering.

### 3.2. Temperate Forest

Figure 5 below shows the radiometric histograms for temperate forest over the greater Germany/Czech Republic region for the strong (top row) and weak (bottom row) beams. Contrary to the distributions observed for boreal forests, temperate forests indicated a visible separability between the ground and canopy distributions with a higher radiometric response attributed to canopy photons rather than ground photons. For temperate forests, the strong beam ground radiometry mode was centered near 0.3 photon/shot, whereas the canopy radiometry mode was centered at approximately 1 photon/shot. Similar to the boreal forest, the night acquisitions in the strong beam showed a higher total radiometry than day acquisitions with "cloud-day" having the lowest distribution of photons. Moreover, what was also consistent with the boreal forest distributions was the clear separability between the day and night acquisitions in the weak beams, which resulted in the night total radiometric values being nearly double those observed during the day. The observed weak beam radiometry in the night acquisitions was actually higher than the observed strong beam radiometry during the day.

### 3.3. Tropical Forest

Figure 6 below shows the radiometric histograms for tropical forest over the Tapajos, Brazil region for the strong (top row) and weak (bottom row) beams. In this situation, for both the strong and weak beam cases, the ground radiometry mode was centered near 0.1 photon/shot, whereas the majority of detected photons were reflected from the canopy. This result was attributed to the high canopy cover observed in tropical forest and less opportunity to capture terrain photons due to obstruction of laser penetration. Similar to both boreal and temperate forests, the night acquisitions in the strong beam showed a higher total radiometry than day acquisitions. Furthermore, when comparing against boreal and temperate forests, there was a clear separability between day and night acquisitions in the weak beams with the night total radiometric values being double those

observed during the day. The weak beam radiometry for night acquisitions was higher than the strong beam day acquisitions.

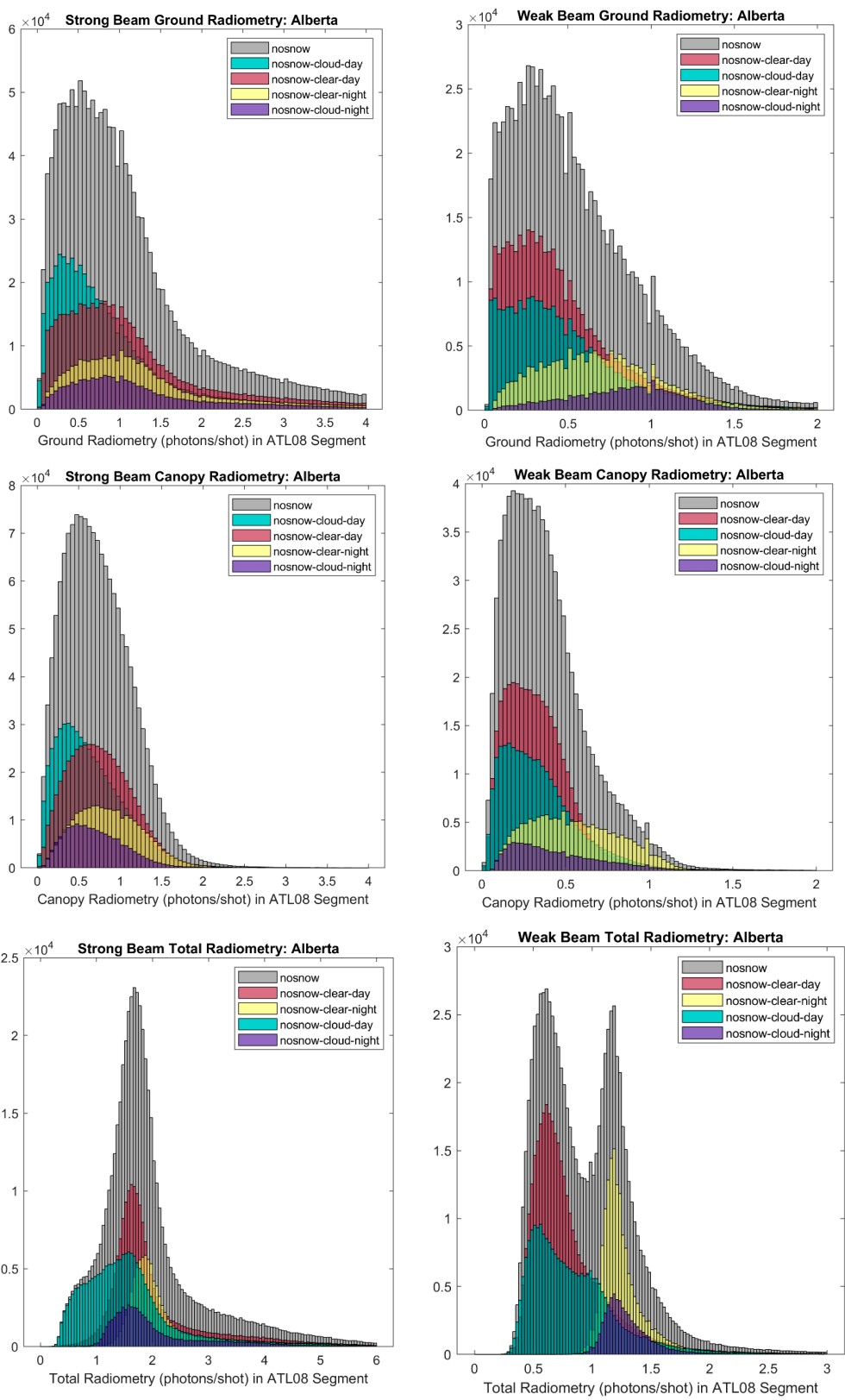

**Figure 4.** Radiometric histograms for ground (**top**), canopy (**center**), and total (**bottom**) from the Alberta data set representing boreal forest in no snow conditions. The histograms for the strong beams are shown on the left and the weak beams are shown on the right.

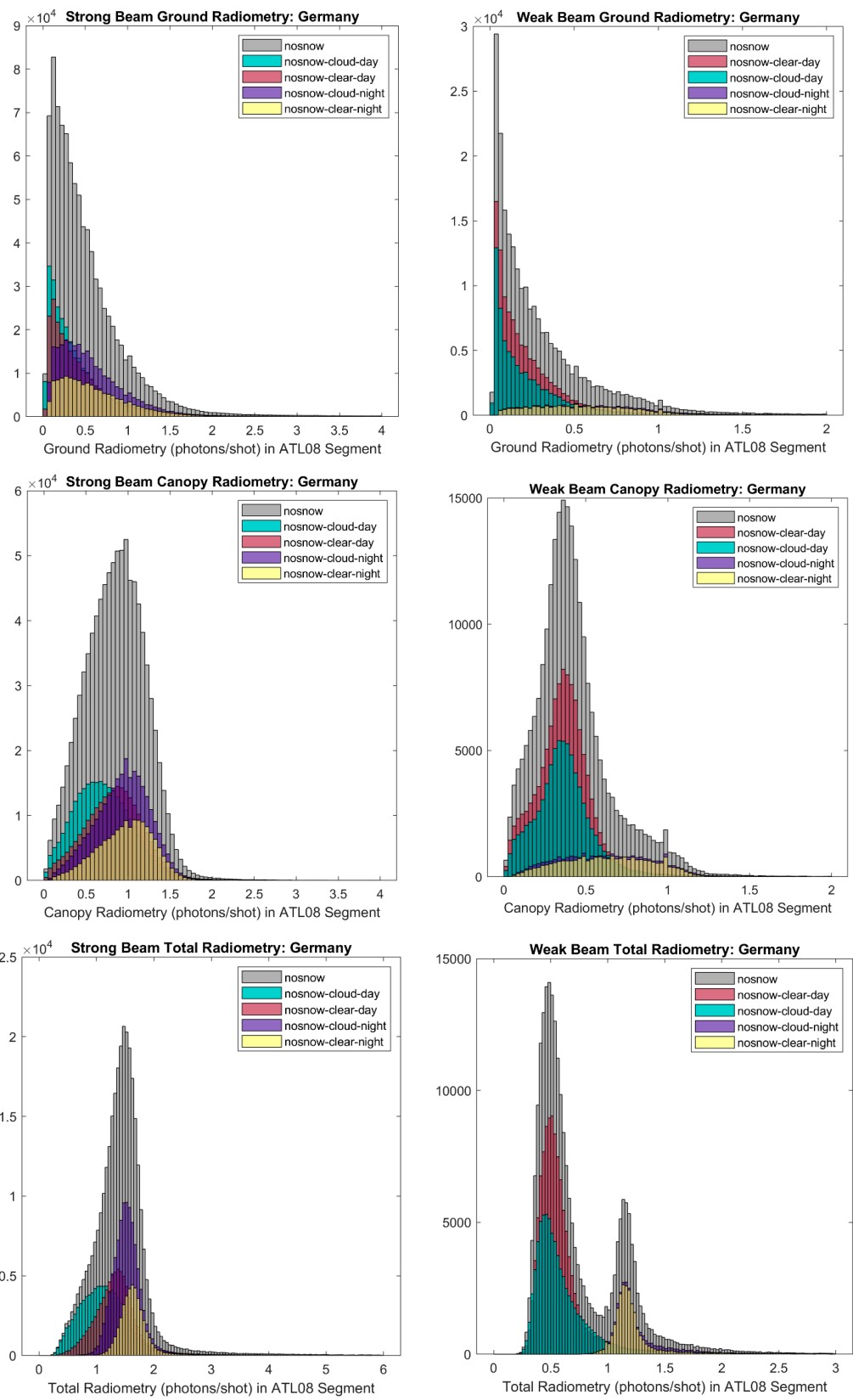

**Figure 5.** Radiometric histograms for ground (**top**), canopy (**center**) and total (**bottom**) from the Germany/Czech Republic data set representing temperate forest in non-snow conditions. The histograms for the strong beams are shown on the left and the weak beams are shown on the right.

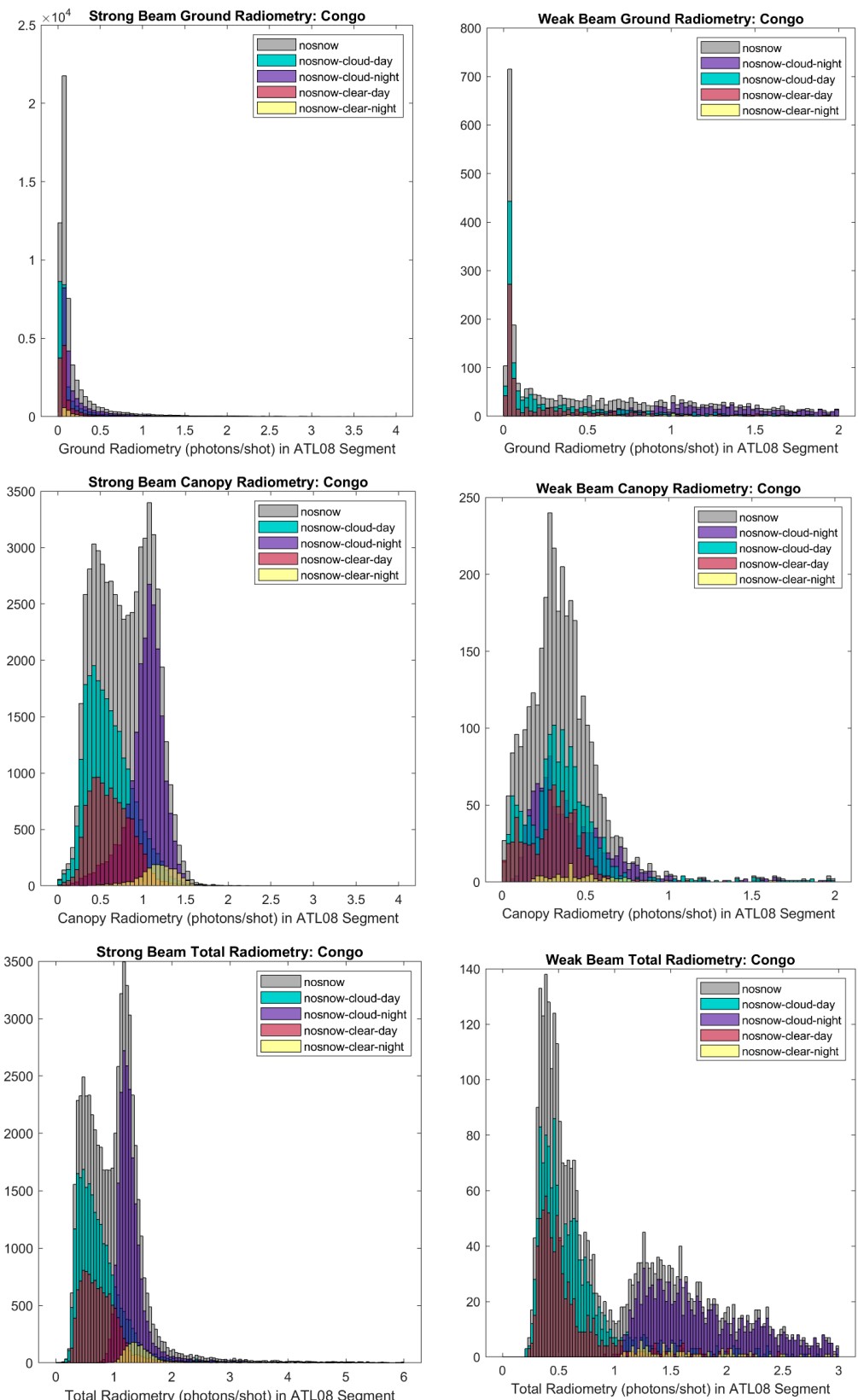

**Figure 6.** Radiometric histograms for ground (**top**), canopy (**center**), and total (**bottom**) from the Tapajos data set representing tropical forest. The histograms for the strong beams are shown on the left and the weak beams are shown on the right.

In summary, the mean total radiometry values for each of these forest types during no-snow conditions (stratified by day versus night and atmospheric scattering), as observed by ICESat-2, are reported in Table 2. Based on this analysis, night acquisitions reigned supreme over day acquisitions for all forest types, as well as all scattering conditions for both the strong and weak beams. With no observed scattering (msw = 0), the strong beam day radiometry performance was approximately 80–88% the night acquisitions. With scattering (msw > 0), the strong beam day radiometry was reduced further to approximately 65–70% of the clear night acquisitions. Atmospheric scattering reduced the strong beam mean total radiometry approximately 20% and 10% for day and night, respectively. For the weak beams, the day radiometry with no observed scattering was approximately 36–66% that of the night acquisitions. When atmospheric scattering was considered, the mean signal radiometry values in the weak beam slightly increased for both day and night acquisitions for reasons unknown.

**Table 2.** Mean total radiometry during no-snow conditions over forest observed by ICESat-2/ATL08 for each study area.

| Region, Forest Type | Strong Beams | | | | Weak Beams | | | |
|---|---|---|---|---|---|---|---|---|
| | MSW = 0, Day | MSW = 0, Night | MSW > 0, Day | MSW > 0, Night | MSW = 0, Day | MSW = 0, Night | MSW > 0, Day | MSW > 0, Night |
| Alberta, Boreal | 2.06 | 2.31 | 1.63 | 2.06 | 0.86 | 1.29 | 0.93 | 1.32 |
| Finland, Boreal | 1.65 | 1.79 | 1.36 | 1.68 | 0.81 | 1.35 | 0.83 | 1.37 |
| Germany, Temperate | 1.38 | 1.63 | 1.19 | 1.56 | 0.66 | 1.27 | 0.70 | 1.33 |
| USA, Temperate | 1.45 | 1.68 | 1.22 | 1.53 | 0.63 | 1.30 | 0.67 | 1.43 |
| Brazil, Tropical | 1.25 | 1.58 | 1.05 | 1.54 | 0.69 | 1.93 | 0.67 | 1.99 |
| Congo, Tropical | 0.89 | 1.62 | 0.88 | 1.32 | 0.75 | 1.67 | 1.01 | 1.84 |

Overall, day acquisitions with atmospheric scattering showed the lowest radiometric response in the strong beam. The boreal forest produced the highest radiometric performance (1.34 to 2.31 photons/shot in strong beams) followed closely by the temperate (1.14 to 1.92 photons/shot in strong beams) and tropical forests (0.77 to 1.68 photons/shot in strong beams). The radiometry for the weak beam cases also indicate that night acquisitions are nearly double the photon returns of day acquisitions. For tropical forest, the night acquisitions from the weak beam are on par or better than both the day and night acquisitions on the strong beams.

## 4. Discussion

The motivation of this research was to quantify the on-orbit radiometric performance of ATLAS over three general forest types: boreal, temperate, and tropical. One of the key findings from our results is that the signal radiometry for night acquisitions is higher than day acquisitions, which is rather unexpected because they should be equivalent. We assumed equivalency was based on the fact that after accounting for solar background noise, the surface reflectance and outgoing laser energy is not changing. This discrepancy is particularly noticeable on the weak beams where the difference is nearly two-fold. In boreal and temperate forests, the day/night difference is 10–15% (strong beam) and 33–51% (weak beam). While it is true that the background noise is reduced in night acquisitions; that should not have an impact on the amount of signal photons reflected from the surface. It is expected that the signal radiometry would remain consistent regardless of solar elevation. This phenomenon, however, is also observed over Antarctica ice sheets. In [10], they reported an approximate ~10% (strong beam) and ~40% (weak beam) reduction in signal strength during the Antarctica summer months.

Prior to launch, predictions of radiometry rates were used to simulate ICESat-2 signal photons and develop algorithms optimized for elevation retrievals over specific surface

types. Table 3 lists the mean observed total radiometric values for the strong and weak beams in snow-free conditions for each of our study areas as well as the pre-launch predictions. Compared to the pre-launch energy level expectation, our two boreal forest cases saw radiometric rates approximately 60–95% more than what was anticipated from the strong beams and the on-orbit weak beams demonstrate roughly four-fold the anticipated value. The observed radiometry for temperate forest strong beam response is slightly lower than the pre-launch prediction (~0.5 a photon), however, the weak beam response is almost double the predicted value. What is rather unexpected is the observed radiometry over tropical forest. Here, the on-orbit radiometry is two to six times higher than the pre-launch estimates for the strong and weak beam, respectively. These results over tropical forest are particularly encouraging as prior to launch the mission was not anticipating quality ATL08 heights over tropical forests due to the expected low signal response. Although this study is only looking at the observed radiometry and not the quality of the extracted heights, it is hypothesized that more photons detected from the surface will improve the likelihood that we can estimate accurate heights from that surface.

**Table 3.** Mean total observed radiometry over three forest types for snow-free conditions.

| Region, Forest Type | On-Orbit Observed | | Pre-Launch Design | |
|---|---|---|---|---|
| | **Strong** | **Weak** | **Strong** | **Weak** |
| Alberta, Boreal | 1.95 | 1.00 | 1.00 | 0.25 |
| Finland, Boreal | 1.63 | 0.87 | 1.00 | 0.25 |
| Germany, Temperate | 1.42 | 0.81 | 1.90 | 0.48 |
| USA, Temperate | 1.49 | 0.83 | 1.90 | 0.48 |
| Tapajos, Tropical | 1.31 | 0.96 | 0.60 | 0.15 |
| Congo, Tropical | 1.05 | 0.89 | 0.60 | 0.15 |

*4.1. ATLAS Per-Beam Radiometric Response*

Another objective of this research was to quantify the radiometric response as a function of ATLAS beam number. Furthermore, we sought to explore the energy ratio between each beam pair. Beam 1 was adjacent to beam 2; beam 3 was adjacent to beam 4; and beam 5 was adjacent to beam 6. For our analysis, we identified and matched the ATL08 segments from the strong beam to an ATL08 weak beam segment. As a reminder, the distance between the weak and strong beam in the cross-track direction is 90 m; thus, each ATL08 segment matched pair is roughly 90 m apart. For our criteria, we used a distance threshold of 100 m. If the distance between a matched pair exceeded our criteria, we rejected those ATL08 segments. Once matched, the ratio between the strong and weak beams were calculated for each pair and then averaged for all ATL08 segments. By only considering matched beam pair segments, we increased the likelihood that each beam pair were observing the same landcover under the same atmospheric scattering conditions.

The mean total radiometry for no snow conditions was calculated for each beam for each of our study areas and is listed in Table 4. The ratio between each beam pair is also listed in Table 4. Among the strong beams, beam 1 consistently yielded the highest radiometric rate followed closely by beam 5. Beam 3 had a consistently 10–15% lower radiometric response than beam 1 for all but one of our study areas in no-snow conditions (Congo was the outlier). The underperformance of beam 3 confirms similar observations reported by [10]. Among the weak beams, beam 4 consistently outperformed beams 2 and 6, which was a pattern not reported in [10]. The ATLAS beam ratio (strong:weak) was designed to be 4:1 based on pre-launch sensor design and energy level analysis. In non-snow conditions, we observed the beam radiometric ratios (strong:weak) range from 1.59 to 2.76; far below the expected 4:1 detection ratio predicted between the strong and weak beam. All things being equal (atmospheric scattering and landcover) and with the spacing between each strong-weak beam pair less than 100 m, it was curious that the beam

ratio ranged from 1.59 to 2.76 when the outgoing energy difference was a factor of 4 and the strong beams had 4 times the available pixels in the detector array compared to the weak beams. These findings contrasted with those observed by [10] where a 4:1 detection ratio was observed over Antarctica ice sheets. Perhaps for lower reflective targets such as forested ecosystems, the ATLAS detector was not fully utilized; however, this still needs further investigation.

**Table 4.** The mean total radiometry by beam on matched segments during no-snow conditions observed by ICESat-2/ATL08 for our study areas.

| Region, Forest Type | Beam 1 | Beam 3 | Beam 5 | Beam 2 | Beam 4 | Beam 6 | B1:B2 | B3:B4 | B5:B6 |
|---|---|---|---|---|---|---|---|---|---|
| Alberta, Boreal | 2.33 | 1.95 | 2.27 | 0.96 | 1.05 | 0.93 | 2.62 | 2.14 | 2.62 |
| Finland, Boreal | 1.86 | 1.61 | 1.85 | 0.76 | 1.00 | 0.81 | 2.76 | 2.09 | 2.68 |
| Germany, Temperate | 1.65 | 1.41 | 1.63 | 0.75 | 0.82 | 0.75 | 2.48 | 2.07 | 2.32 |
| USA, Temperate | 1.69 | 1.42 | 1.65 | 0.75 | 0.81 | 0.77 | 2.67 | 2.18 | 2.60 |
| Brazil, Tropical | 1.52 | 1.38 | 1.54 | 0.70 | 1.20 | 0.78 | 2.57 | 1.84 | 2.50 |
| Congo, Tropical | 1.16 | 1.22 | 1.33 | 0.65 | 1.08 | 0.70 | 2.12 | 1.59 | 2.22 |

Table 5 reports the mean total radiometry values for each beam when snow was identified in the ATL08 segments. Given the increase in reflectance for snow covered surfaces from non-snow conditions the radiometric response, as expected, also increased. The mean value of the total radiometry ranges from 1.5 to 2.5 times more than what was observed for non-snow conditions. With snow on the landscape, the ratio between the strong to weak beam also increased by approximately 30–40%. However, the ratio between the strong and weak beams was still less than the expected value of 4.

**Table 5.** The mean total radiometry by beam on matched segments during snow conditions observed by ICESat-2/ATL08 for our study areas.

| Region, Forest Type | Beam 1 | Beam 3 | Beam 5 | Beam 2 | Beam 4 | Beam 6 | B1:B2 | B3:B4 | B5:B6 |
|---|---|---|---|---|---|---|---|---|---|
| Alberta, Boreal | 4.95 | 3.82 | 4.82 | 1.46 | 1.42 | 1.45 | 3.40 | 2.72 | 3.34 |
| Finland, Boreal | 4.54 | 3.53 | 4.41 | 1.35 | 1.33 | 1.35 | 3.39 | 2.71 | 3.29 |
| Germany, Temperate | 3.62 | 2.93 | 3.67 | 1.16 | 1.13 | 1.17 | 3.29 | 2.72 | 3.26 |
| USA, Temperate | 3.47 | 2.54 | 3.37 | 1.08 | 1.02 | 1.07 | 3.43 | 2.63 | 3.32 |
| Brazil, Tropical | N/A | N/A | N/A | N/A | N/A | N/A | N/A | N/A | N/A |
| Congo, Tropical | N/A | N/A | N/A | N/A | N/A | N/A | N/A | N/A | N/A |

### 4.2. Radiometric Profiles

Because ICESat-2 is a photon counting system, rather than a full-waveform system, only a few photons are detected by the ATLAS receiver onboard ICESat-2. As mentioned previously, the number of detected photon events associated with each outgoing laser pulse was a function of the transmitted laser energy, surface reflectance, solar conditions, and scattering and attenuation in the atmosphere. The number of photons detected, as well as where within the canopy they were reflected, was a function of the reflectance of both the ground and canopy [5]. Over non-snow-covered land, approximately 1–2 photons reflected per outgoing shot was observed in the strong beam ICESat-2 data. As illustrated in Table 5, the presence of snow on the surface increased the number of detected photons due to the increased reflectance of snow at 532 nm.

Canopy cover—a measure of vegetation structure—is an integral descriptor when characterizing a forested environment with remotely sensed data. Canopy cover is the layer formed by the branches and crowns of plants or trees. With lidar data, canopy cover is

often represented as the ratio of canopy returns to the total number of returns (i.e., canopy and ground). Canopy cover is not currently included on the ATL08 data product due to varying surface reflectance values. Figure 7 is an example of how the changing surface reflectance can change the number of reflected photons. In this illustration, we considered the same plot of land with the same trees and the only difference was the presence of snow. If canopy cover was calculated as a simple ratio of canopy photons to total photons, the canopy cover estimated changes in this example based solely on the reflectance and not the actual vegetation structure within each plot. In this simple case, our no snow (left) representation resulted in 1 detected photon from the ground and 1 detected photon from the canopy; yielding a canopy cover estimate of 50%. In the snow-covered ground case (middle), it was feasible that there would be three detected photons from the ground and only 1 photon from the canopy, resulting in a canopy cover estimate of 25%. In the canopy and ground snow covered case (right), it was conceivable that we could see five photons detected from the ground and three photons from the canopy, resulting in a canopy cover estimate of 38%. While these numbers are notional, they provide an illustration regarding the subtlety of photon counting and vegetation structure.

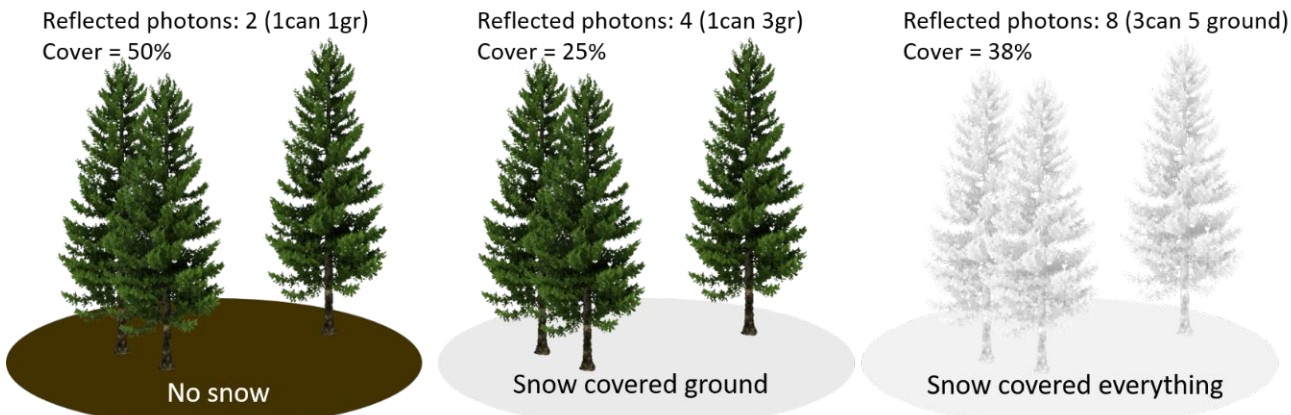

**Figure 7.** Illustration highlighting the role of reflectance for canopy cover determination.

As shown in the literature, canopy cover can be directly obtained from lidar data [17–20]. For a full waveform system, the integrated backscatter component for the ground and vegetation, $R_g$ and $R_v$ are a function of the total waveform energy. ICESat-2 is photon sampling system leaving $R_g$ and $R_v$ unknown; instead we can substitute the number of detected photons for both the ground and canopy. The number of detected photons $R_g$ and $R_v$ are a function of the atmospheric transmission and surface reflectance. However, we hypothesize that $R_g$ and $R_v$ the along-track segment are a function of the ground reflectance, $\rho_g$, and the vegetation reflectance, $\rho_v$, weighted by the amount of vegetation cover as we graphically depicted in Figure 7. The general equation for retrieving canopy cover from waveform lidar data as presented in Armston et al. (2013) is shown in Equation (2).

$$P_{gap}(z) = 1 - \frac{\sum_{z=z_i}^{z=max(z)} R_v}{R_v} \frac{1}{1 + \frac{\rho_v}{\rho_g} \frac{R_g}{R_v}} \tag{2}$$

A critical piece of information needed to solve Equation (2) in a direct way is an estimate of $\rho_v$ and $\rho_g$. Through the analysis of partitioning between ground and canopy radiometry is the likely path for calculating the vegetation structure within each ATL08 segment. The relationship of the ground radiometry against the canopy radiometry creates the radiometric profile for that region. Figure 8 presents the radiometric profiles for Beam 1 for Finland, Germany, and Tapajos (top row) and Alberta, USA, and Congo (bottom row) during optimal conditions. That is, each of these radiometric profiles are for clear atmospheric conditions and at night. Since the night cases had the highest radiometric

values in all situations, the radiometric profiles are generated for those ATL08 segments. What is evident in each of the radiometric profiles is a clustering of radiometric values that form an approximate linear trend: a green line was manually inserted to highlight the trend. This linear relationship, i.e., the distance along the line, provides the canopy cover and we refer to it here as the canopy cover line. That is, the y-intercept corresponds to ATL08 segments having a 100% cover and the x-intercept corresponds to ATL08 segments having 0% canopy cover. For example, in the cases of tropical forest most of those points are clustered near the y-intercept indicating a high canopy cover value. This canopy cover line is analogous to the cover line presented in [18] for full waveform lidar data.

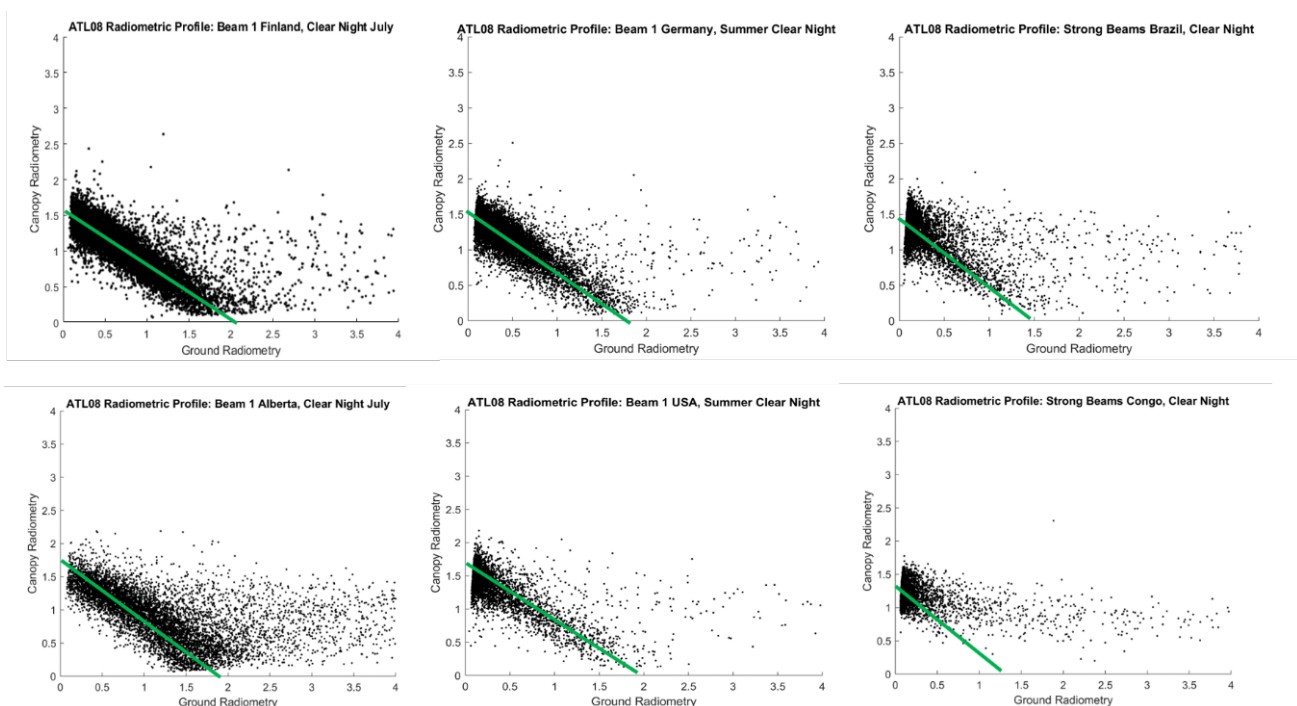

**Figure 8.** Scatter plots between the ground radiometry and canopy radiometry for each of the six study locations. These plots represent the observed ATL08 segment radiometric profile.

Although the majority of points are situated around the canopy cover line, there are a few outliers with a high ground radiometry value in all of these examples. The intercept values in the radiometric profiles are directly related to $\rho_v$ and $\rho_g$ at that given epoch and area. Each ATL08 segment used in this analysis is selected as a forest class from the ATL08 landcover classification. The landcover values written to ATL08 (release 004) are derived from the coarser resolution MODIS product. Thus, the landcover classification on ATL08 does not account for heterogeneity of the vegetation. By knowing the ground surface reflectance, $\rho_g$ and vegetation reflectance, $\rho_v$ for a given latitude and longitude, it is possible to determine the canopy cover based on the ICESat-2 radiometry, that is the number of detected photons per outgoing laser pulse. Determination of canopy cover via this method is currently under development.

### 4.3. NOAA Snow Flag

As evidenced from the basic findings shown here, the presence of snow on the landscape makes a significant impact on the number of photons detected. As such, the inclusion of the NOAA snow flag on the data product is a good indicator regarding the presence of snow on the landscape. NOAA daily snow cover product [21] is reported at 1 km resolution and thus all ATL08 segments that fall within the 1 km resolution will carry that flag value. Figure 9 depicts the radiometric profile of individual ATL08 segments over Finland for non-snow forests, night, and clear skies in the month of November (left panel) and for

snow-covered forest (right panel). As a reminder, these monthly profiles include data from multiple epochs over a two-year period. As evident in the non-snow-covered panel, there are many ATL08 segments with radiometric values higher than bulk of segments clustered around the canopy cover line. We hypothesize that these high values are higher due to snow increasing the reflectance within that ATL08 segment, but they are not flagged as snow-covered in the NOAA daily snow cover product. Similarly, we assume that there are ATL08 segments that are flagged as being snow covered are actually not snow covered. Again, this is a function of the coarse resolution (1 km) of the NOAA product applied to all ATL08 segments at 100 m beneath it. As such, we recommend that if users are incorporating the NOAA snow flag into their analysis as a means for filtering the data that they recognize that there may be discrepancies.

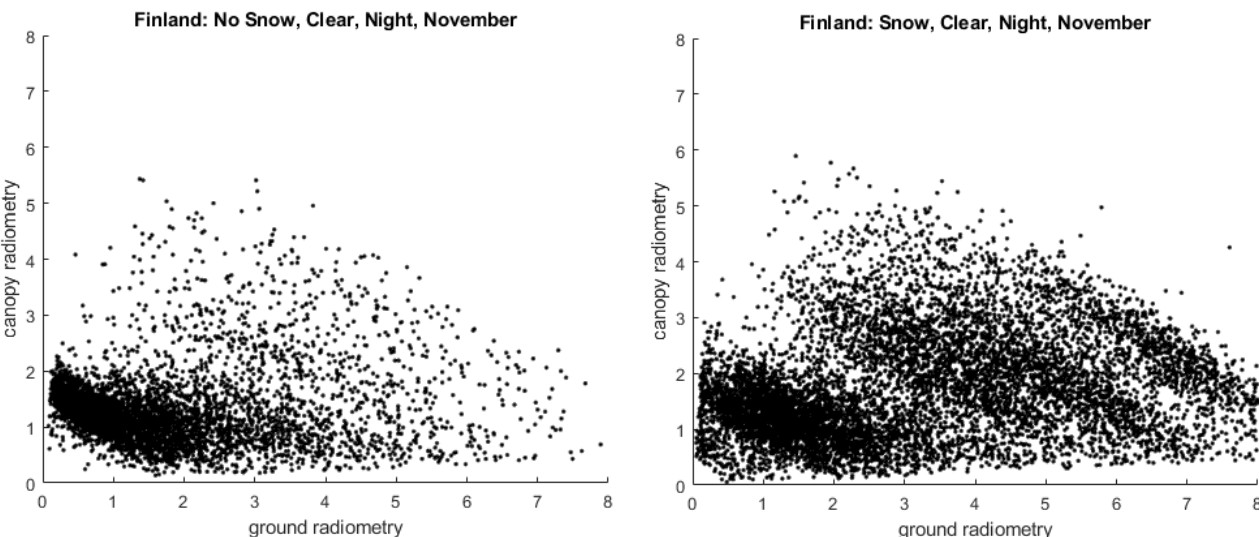

**Figure 9.** Radiometric profile for Finland for non-snow conditions during November (**left**) and for snow conditions (**right**).

As refinements to the ATL08 data product continue throughout the mission, we anticipate that the radiometric characterization of forested ecosystems will continue to improve. Release 005 of the ATL08 data product will incorporate the landcover value derived from the 2019 Copernicus land cover product at the 100 m resolution. This change from the MODIS landcover to the Copernicus land cover is a better match spatially to ATL08 and it is also temporally closer to the ICESat-2 data. Additional modifications to the ATL08 data product that are currently under development by the ATL08 data product team include the correct calculation of the canopy radiometric value on the ATL08 data product as well as improved ground finding. It is anticipated that future versions of the ATL08 data product will include an ICESat-2 derived estimate of snow cover at the 100 m resolution of the data product.

## 5. Conclusions

The goal of this work was to establish a baseline understanding of the radiometric performance of the ATLAS sensor onboard ICESat-2 over three broad forest types. Through this analysis, we observed that ATLAS strong beam radiometry exceeded the pre-launch design cases for boreal and tropical forests but underestimated the predicted radiometry over temperate forests by approximately half a photon. The weak beams, in contrast, exceeded all pre-launch conditions by a factor of two to six for all forest types. We also observed that the signal radiometry from day acquisitions was lower than night acquisitions by 10% and 40% for the strong and weak beams, respectively. This pattern was also observed and reported by [10]. We found that beam 3 (center ground track, strong beam) had approximately 10% lower radiometry than the other strong beams. This pattern of

beam strength amongst the strong beams was confirmed in other analyses over more reflective ice sheets [9,10]. Another finding from this analysis was the ratio between the strong and weak beams, which ranged from ~2:1 to 2.7:1 for non-snow forested landscapes rather than the expected 4:1 ratio. When ATL08 segments were snow covered, this ratio increased to ~3.25:1. This analysis also showed that acquisitions acquired during the day with atmospheric scattering saw the lowest radiometric values. In several instances, the radiometric response of the weak beams at night was found to be similar to the strong beams acquired during day. The results from this study were intended to be informative and perhaps serve as a benchmark for filtering or analysis of the ATL08 data products. Although certain beams or acquisition scenarios may be deemed superior radiometrically, a higher number of photons does not always imply correct terrain or canopy heights. The photons detected here were the starting point but still required accurate ground and canopy finding from the ATL08 algorithm, although the likelihood of correctly labeling the ground and canopy should increase with more photons.

**Author Contributions:** Conceptualization, A.N.; methodology, A.N.; software, A.N. and E.G.; validation and formal analysis, A.N.; investigation, A.N. and E.G.; resources, A.N.; writing—original draft preparation, A.N.; writing—review and editing, A.N., L.M., S.H. and M.P.; funding acquisition, A.N. All authors have read and agreed to the published version of the manuscript.

**Funding:** This research was funded by NASA grant 80NSSC20K0965 (PI Neuenschwander).

**Institutional Review Board Statement:** Not applicable.

**Informed Consent Statement:** Not applicable.

**Data Availability Statement:** All data used for this manuscript are openly available from the National Snow and Ice Data Center https://nsidc.org/data/icesat-2 (accessed on 8 December 2021).

**Acknowledgments:** The authors wish to thank the NASA ICESat-2 Science Team and Project Science Office for their hard work preparing the data used in this study. We would also like to acknowledge our research team, especially Mike Alonzo at the University of Texas at Austin for helping with data retrievals from NSIDC. All ICESat-2 data are publicly available through the National Snow and Ice Data Center (https://nsidc.org/data/icesat-2) (accessed on 8 December 2021).

**Conflicts of Interest:** The authors declare no conflict of interest.

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
