# Peer review of "Radiometric Assessment of ICESat-2 over Vegetated Surfaces"

_remotesensing, doi:10.3390/rs14030787_

Round 1

Reviewer 1 Report

The paper studies and quntifies the on-orbit radiometric performance of ALTAS data over boreal, temperate and tropical forests. The study has some novelty for sure. However, the organization and writing need to be improved. E.g., the abstract is too long and mentions too much specific details and results.

Line 14: The fullform of ATLAS is not provided in the text.

Line 270: Figure 4, 5: The title needs to be refined. Please clarify which is the attribute value used in forming the histogram?

Line 271: Also, please clarify why was sub-tropical forest class not analyzed in the experiments?

Line 289: Please note that the legends in most figures are not readable.

Line 338: The authors point out that the increase in signal radiometry is unexpectedly high during the night? Isn't it a common phenomen with most senors as during the day the sun contributes more to background noise (e.g., in the 532 nm too)?

Line 406: Why is the total radiometry associated with Brazil and Congo empty?

Line 561: The number of recent papers are very-low in the references. More recent works need to be studied and cited.

Author Response

Reviewer #1  
Comments and Suggestions for Authors
The paper studies and quntifies the on-orbit radiometric performance of ALTAS data over boreal, temperate and tropical forests. The study has some novelty for sure. However, the organization and writing need to be improved. E.g., the abstract is too long and mentions too much specific details and results.
Thank you for your comments and suggestions. Based on your suggestion, we did shorten the length of the abstract to 249 words and removed a few of the details reported in the results section of the text. In addition, based on your general concern about organization we reviewed the structure and flow of the content to support optimal reader interpretation as best as we could. 

Line 14: The fullform of ATLAS is not provided in the text.
Line 14: Although provided in the first paragraph of the main text, we added the full form of ATLAS in the abstract.

Line 270: Figure 4, 5: The title needs to be refined. Please clarify which is the attribute value used in forming the histogram?

Figures 4 -6: We added a title to each set of histograms to add clarity. The attribute that is used to create the histograms are listed on the x-axis of each histogram. We did however, clarify this concept in the text. The figures are 300dpi resolution, so we could lay them out differently in the manuscript to make them easier to read if that is something the publisher wishes to do. 

Line 271: Also, please clarify why was sub-tropical forest class not analyzed in the experiments?
For this paper, our focus was to examine the forest classes of boreal, temperate, and tropical to maintain consistency with the vegetation ATLAS signal design cases for the mission prior to launch.  Sub-tropical was not a pre-launch design case for the mission –thus we did not include it in our analysis.

Line 289: Please note that the legends in most figures are not readable.
We have addressed the resolution of the figures such that the labels are easier to read. The resolution of the figures are each 300 dpi, however, if the publisher wishes to change the layout so that they are more legible (i.e. larger) we would certainly accommodate that request. We will keep this in mind during the publication process once/if we get to the final copy review.

Line 338: The authors point out that the increase in signal radiometry is unexpectedly high during the night? Isn't it a common phenomen with most senors as during the day the sun contributes more to background noise (e.g., in the 532 nm too)?
Line 338: responding to the question regarding the different signal radiometry rates between day and night, yes…the background rate is higher during the day and the total photon rate would be higher during the day. However we would still expect that the signal rate (that is, the actual number of photons reflecting from the surface) would be similar to night acquisitions. 

Line 406: Why is the total radiometry associated with Brazil and Congo empty?
Line 406: Table 5 depicts the radiometry in snow conditions. Tropical forest in Brazil and Congo do not have snow, thus those values are empty in the Table.  The table entry has been changed to N/A rather than the – empty notation.

Line 561: The number of recent papers are very-low in the references. More recent works need to be studied and cited.

We agree, the number of papers included in the references are low. To alleviate this issue we have included 7 relevant citations to support various aspects in the manuscript. If the reviewer has any specific references in mind that are not represented in the paper, we would be glad to add them. 

Reviewer 2 Report

reviewer #1: General comments

The paper presents an interesting case study of The Ice, Cloud and Land Elevation Satellite-2 (ICESat-2) data to explore the radiometric performance of the ATLAS 80 instrument over different forest types. The general approach is innovative in terms of RS techniques and methods. Authors initiate an interesting study about the use of ICESat-2 to determine the influence of radiometry on the estimated terrain and canopy heights from the ATL08 (land and vegetation) data product. As the author mentioned, the proposed methods for filtering or analysis of the ATL08 data products over vegetated surfaces.in the literature are rough or scare using global forest height product. This is a good systematic framework for filtering or analysis of the ATL08 data products over vegetated surfaces. The introduction is comprehensive and well written. The method and results are also well described in the manuscript. The full scope of the types of forest where ICESat-2 products that can be used as an alternative to ALS data also continues to be an active area of research. The manuscript can be considered a valuable contribution to the literature and should be considered for publishing. However, some minor corrections should be considered and done.

Minor issues

Specific comments

Introduction

L57. I can not see the references Magruder et al., 2020 in the list of references.

Material and methods

L90. I can not see the references of (Neumann et al., 2019) in the list of references and also of (Magruder et al., 2021) in L96.

Author Response

Reviewer #2
The paper presents an interesting case study of The Ice, Cloud and Land Elevation Satellite-2 (ICESat-2) data to explore the radiometric performance of the ATLAS 80 instrument over different forest types. The general approach is innovative in terms of RS techniques and methods. Authors initiate an interesting study about the use of ICESat-2 to determine the influence of radiometry on the estimated terrain and canopy heights from the ATL08 (land and vegetation) data product. As the author mentioned, the proposed methods for filtering or analysis of the ATL08 data products over vegetated surfaces.in the literature are rough or scare using global forest height product. This is a good systematic framework for filtering or analysis of the ATL08 data products over vegetated surfaces. The introduction is comprehensive and well written. The method and results are also well described in the manuscript. The full scope of the types of forest where ICESat-2 products that can be used as an alternative to ALS data also continues to be an active area of research. The manuscript can be considered a valuable contribution to the literature and should be considered for publishing. However, some minor corrections should be considered and done.

Minor issues

Specific comments

Introduction

L57. I can not see the references Magruder et al., 2020 in the list of references.

Material and methods

L90. I can not see the references of (Neumann et al., 2019) in the list of references and also of (Magruder et al., 2021) in L96.

The references mentioned in the text and highlighted by reviewer #2 are now included in the reference section.

Round 2

Reviewer 1 Report

The manuscript has greatly improved and is now in good shape. However, there are some concerns that need to be addressed:

Line 42: What is the weak beam detector size? It reads like detector size is mentioned only for strong beams.

Line 56: The authors themselves have mentioned that the number of detected photons is a function of solar conditions. However, the response to review question 6 (i.e., the one corresponding to line 338) says that "one would still expect that the signal rate (that is, the actual number of photons reflecting from the surface) would be similar to night acquisitions." These are contradictory statements, and hence need clarification! Also, In Line 65, the authors have claimed to have proved it already in an earlier paper. In any case, he statement should be like " Our observation is in alignment with the expected outcome..." (rather than depicting it is as a surprise).

LIne 220: "Here, N from Equation 1 is" can be replaced by "where N is

line2 538,  539: Make sure the ratios are exact and not an approximation. It can be seen that at some places they are exact while at other places they are approximated.

Line 529 : Please correct the typo "photonThe weak".  FYI: There are typos/grammatical issues in many places.
